# Long-Term Effects of a Stepwise, Multimodal, Non-Restrictive Antimicrobial Stewardship Programme for Reducing Broad-Spectrum Antibiotic Use in the ICU

**DOI:** 10.3390/antibiotics13020132

**Published:** 2024-01-29

**Authors:** Mar Ronda, Victor Daniel Gumucio-Sanguino, Evelyn Shaw, Rosa Granada, Fe Tubau, Eva Santafosta, Joan Sabater, Francisco Esteve, Cristian Tebé, Rafael Mañez, Jordi Carratalà, Mireia Puig-Asensio, Sara Cobo-Sacristán, Ariadna Padullés

**Affiliations:** 1Department of Infectious Diseases, Hospital Universitari de Bellvitge, Feixa Llarga s/n, 08907 L’Hospitalet de Llobregat, Barcelona, Spain; mronse1@gmail.com (M.R.);; 2Department of Intensive Care, Hospital Universitari de Bellvitge, Feixa Llarga s/n, 08907 L’Hospitalet de Llobregat, Barcelona, Spain; vgumucio@bellvitgehospital.cat (V.D.G.-S.);; 3Infectious Diseases and Transplantation Division, Institut d’Investigació Biomèdica de Bellvitge (IDIBELL), Av. Gran Via de l’Hospitalet 199, 08908 L’Hospitalet de Llobregat, Barcelona, Spain; 4Centro de Investigación Biomédica en Red de Enfermedades Infecciosas (CIBERINFEC), Instituto de Salud Carlos III, Av. Monforte de Lemos, 3-5, 28029 Madrid, Spain; apadulles@bellvitgehospital.cat; 5Department of Microbiology, Hospital Universitari de Bellvitge, Feixa Llarga s/n, 08907 L’Hospitalet de Llobregat, Barcelona, Spain; 6Centro de Investigación Biomédica en Red de Enfermedades Respiratorias (CIBERES), Instituto de Salud Carlos III, Av. Monforte de Lemos, 3-5, 28029 Madrid, Spain; 7Biostatistics Support and Research Unit, Germans Trias i Pujol Research Institute and Hospital (IGTP), Carretera de Can Ruti, Camí de les Escoles s/n, 08916 Badalona, Barcelona, Spain; 8Department of Clinical Sciences, Faculty of Medicine and Health Sciences, University of Barcelona, Campus Bellvitge, Feixa Llarga s/n, 08907 L’Hospitalet de Llobregat, Barcelona, Spain; 9Department of Pharmacy, Hospital Universitari de Bellvitge, Feixa Llarga s/n, 08907 L’Hospitalet de Llobregat, Barcelona, Spain; 10Pharmacotherapy, Pharmacogenetics and Pharmaceutical Technology, Institut d’Investigació Biomèdica de Bellvitge (IDIBELL), Av. Gran Via de l’Hospitalet 199, 08908 L’Hospitalet de Llobregat, Barcelona, Spain

**Keywords:** antimicrobial stewardship, interrupted time series analysis, ICU, broad-spectrum antibiotics, appropriateness of antibiotic prescribing

## Abstract

Information on the long-term effects of non-restrictive antimicrobial stewardship (AMS) strategies is scarce. We assessed the effect of a stepwise, multimodal, non-restrictive AMS programme on broad-spectrum antibiotic use in the intensive care unit (ICU) over an 8-year period. Components of the AMS were progressively implemented. Appropriateness of antibiotic prescribing was also assessed by monthly point-prevalence surveys from 2013 onwards. A Poisson regression model was fitted to evaluate trends in the reduction of antibiotic use and in the appropriateness of their prescription. From 2011 to 2019, a total of 12,466 patients were admitted to the ICU. Antibiotic use fell from 185.4 to 141.9 DDD per 100 PD [absolute difference, −43.5 (23%), 95% CI −100.73 to 13.73; *p* = 0.13] and broad-spectrum antibiotic fell from 41.2 to 36.5 [absolute difference, −4.7 (11%), 95% CI −19.58 to 10.18; *p* = 0.5]. Appropriateness of antibiotic prescribing rose by 11% per year [IRR: 0.89, 95% CI 0.80 to 1.00; *p* = 0.048], while broad-spectrum antibiotic use showed a dual trend, rising by 22% until 2015 and then falling by 10% per year since 2016 [IRR: 0.90, 95% CI 0.81 to 0.99; *p* = 0.03]. This stepwise, multimodal, non-restrictive AMS achieved a sustained reduction in broad-spectrum antibiotic use in the ICU and significantly improved appropriateness of antibiotic prescribing.

## 1. Introduction

Antimicrobial resistance (AMR) is a priority issue for the World Health Organization. It is responsible for an estimated 700,000 deaths annually due to hard-to-treat infections [1,2]. Additionally, misuse and overuse of antimicrobial agents in human medicine and food production chains have been identified as main drivers of AMR [3].

Antimicrobial stewardship (AMS) programmes have a key role to play in reducing AMR and tackling the emergence and spread of multidrug-resistant (MDR) bacteria. It is essential to implement effective measures able to reduce unnecessary and inappropriate use of antimicrobial agents without negatively affecting patient outcomes [4,5,6].

At most hospitals, intensive care units (ICUs) are among the highest consumers of broad-spectrum antibiotics. Prompt, appropriate administration of antibiotics is associated with survival benefit in critically ill patients with sepsis. However, once the patient’s clinical status improves, ICU physicians are often reluctant to narrow the antibiotic spectrum in response to the microbiological results, and they are also unwilling to discontinue antibiotics despite that infection has not been confirmed. This frequently leads to the overuse of antibiotics [7,8,9,10].

AMS strategies based on restrictive policies have been associated with rapid reductions in targeted broad-spectrum antibiotic use in ICUs, which is particularly useful in outbreak settings. However, these policies have often been accompanied by increased prescribing of other antibiotics of similar spectrum, thus preventing a reduction in overall broad-spectrum antibiotic use [11,12]. In addition, clinicians do not always accept restrictive policies. In this regard, a Cochrane systematic review of interventions to improve antibiotic prescribing practices found four non-randomised studies that reported negative effects of restrictive policies on professional culture, including breakdowns in trust and communication [13]. 

Non-restrictive AMS policies (also known as persuasive strategies) are the ones that physicians prefer for reducing antibiotic use [14,15]. A recent cross-sectional survey conducted in France, which explored the preferred AMS strategies of hospital prescribers, found that prescribers considered educational actions to be more useful than restrictive ones. They felt that restrictive interventions undermined their clinical autonomy [16]. 

Studies investigating the social and cultural determinants of the prescribing behaviours of physicians in the hospital setting have identified some physician prescribing beliefs, such as that senior doctors often base their decisions on personal knowledge and experience despite not having specific knowledge about antimicrobial use. In addition, junior doctors tend to base their antibiotic prescribing on a senior physician’s decision [16,17,18]. Consequently, involving senior staff in the development of local AMS interventions may promote changes in prescribing habits, especially among younger physicians, and help to address inappropriate antibiotic prescribing. 

AMS programmes in ICUs rarely use only non-restrictive interventions, and those that do often report results obtained only during short follow-up periods [19,20,21,22,23]. Information regarding the long-term effects of non-restrictive AMS strategies on antibiotic use in ICUs is scarce. Our ICU initiated an AMS programme as part of a wider intervention to control endemic MDR bacteria [24]. The programme also included interventions involving senior ICU leaders to promote adherence to guidelines and reduce broad-spectrum antibiotic consumption. The present study assessed the long-term effects of a stepwise, multimodal, non-restrictive AMS programme on broad-spectrum antibiotic use and the appropriateness of antibiotic prescribing in the ICU. 

## 2. Results

A total of 12,466 patients were admitted to the ICU from 2011 to 2019. The mean age of the patients was 63 years (SD14) and the mean APACHE II severity of illness score was 16 (SD 8). See Appendix A.

### 2.1. Antibiotic Use

Antimicrobial consumption, including antibiotic, antifungal, antiviral, and antiparasitic use, fell from 204.1 in 2011 to 164.7 DDD per 100 PD in 2019 [absolute difference, −39.4 (19%), 95% CI −113.54 to 34.74; *p* = 0.18]. The decrease was mainly driven by antibiotic consumption, which fell from 185.4 in 2011 to 141.9 DDD per 100 PD in 2019 [absolute difference, −43.5 (23%), 95% CI −100.73 to 13.73; *p* = 0.13]. The use of broad-spectrum antibiotics fell from 41.2 to 36.5 [absolute difference, −4.7 (11%), 95% CI −19.58 to 10.18; *p* = 0.5]. 

Broad-spectrum antibiotic use in general, and carbapenem use in particular, showed a dual trend. From 2011 to 2015, the use of broad-spectrum antibiotics increased by 9.1 (22%) and that of carbapenems by 8.7 (40%) DDD per 100 PD. However, since 2016, broad-spectrum antibiotic use presented a significant reduction of 10% per year [IRR: 0.90, 95% CI 0.81 to 0.99; *p* = 0.03] and carbapenem use of 16% per year [IRR: 0.84, 95% CI 0.74 to 0.97; *p* = 0.01]. 

Penicillin use fell from 56.2 in 2011 to 39.6 DDD per 100 PD in 2019 [absolute difference, −16.6 (29%); 95% CI −27.85 to −5.35; *p* = 0.01], fluoroquinolones from 15.5 to 13.1 DDD per 100 PD [absolute difference, −2.4 (15%), 95% CI −7.73 to 2.93; *p* = 0.3], colistin from 32.4 to 3.3 DDD per 100 PD [absolute difference, −29.1 (90%), 95% CI −52.06 to −6.14; *p* = 0.07], and aminoglycosides from 3.9 to 1.2 DDD per 100 PD [absolute difference, −2.7 (69%), 95% CI −5.57 to 0.13; *p* = 0.1]. In contrast, cephalosporin use rose over the study period, from 14.4 in 2011 to 16.6 DDD per 100 PD in 2019 [absolute difference, +2.2 (15%), 95% CI −4.12 to 8.52; *p* = 0.4]. See Figure 1 and Figure 2. Appendix A shows the main classes of antibiotics in use in the ICU.

### 2.2. Appropriateness of Prescriptions

The point-prevalence surveys showed that 1838 (71%) out of the 2573 patients surveyed were receiving one or more antimicrobial therapies, resulting in a total of 3098 antimicrobials assessed for appropriateness, of which 90% were antibiotics. The number of patients receiving antibiotics in the monthly surveys did not decrease throughout the study period. There was a trend towards an increase in the number of patients with antibiotic treatment over time. See Figure 3 and Appendix A. 

From 2013 to 2019, adherence to guideline duration of therapy increased from 83.5% to 87.02% [absolute difference +3.5%, 95% CI −1.5 to +8.4, *p* = 0.17], and de-escalation according to culture results rose from 83.5% to 92.9% [absolute difference +9.4%, 95% CI +4.7 to +13.9, *p* < 0.001]. Appropriateness of antibiotic prescribing increased from 75.9% to 85.6% [absolute difference +9.7%, 95% CI +4.1 to +15.1, *p* < 0.001], reflecting a significant increase each year in appropriateness of antibiotic prescribing since the start of the AMS programme [IRR: 0.89, 95% CI: 0.8 to 1.0, *p* = 0.048]. See Figure 4.

### 2.3. Outcomes 

No differences in length of stay, ICU mortality, and 30-day ICU readmissions were observed with the AMS implementation. See Appendix A.

## 3. Discussion

The study shows that an AMS programme based on stepwise, multimodal, non-restrictive interventions lowered antibiotic use in the ICU by 23% over an 8-year period. The programme also significantly reduced the use of broad-spectrum antibiotics by 10% per year from 2016 onwards. Interestingly, the reduction in antibiotic use was due not to a reduction in the number of patients receiving antibiotics over time but to an increase in the appropriateness of antibiotic prescribing.

Few studies have reported long-term reductions in antimicrobial use associated with AMS programmes. Alvarez-Lerma et al. observed a significant reduction in overall antimicrobial use after 5 years of implementing a multimodal non-restrictive AMS programme in a 14-bed ICU. Nevertheless, assessing the effect of the programme on the results in that study was complicated by the fact that the authors concurrently implemented strategies that decreased healthcare-associated infections (HAIs) in their ICU by 50% [25]. In a cohort study in four Canadian ICUs, Morris et al. reported a significant decrease in antibacterial use, especially anti-MRSA and antipseudomonal drugs. Their AMS was based on coaching plus audit and feedback [26]. Adhikari et al. described an AMS programme that comprised restrictive and non-restrictive strategies in a 15-bed ICU in Australia and achieved a sustained reduction in broad-spectrum Gram-negative antibiotic use over a 7-year period. Once again, however, rates of HAIs also fell during the study period, and this factor may have influenced the results [27].

Our study achieved an early reduction in antibiotic use after 1 year of implementation of our AMS programme, in agreement with other studies with short-term follow-ups [20,28,29]. Subsequently, the trend in antibiotic use increased particularly for broad-spectrum antibiotics and the programme did not achieve a steady decline in antibiotic use until 2016. Notably, the sharp reduction in colistin use observed during the study period was directly related to reductions in endemic *Acinetobacter baumanni* and other MDR Gram-negative bacteria (GNB) following the implementation of new cleaning policies and removal of sinks from ICU rooms, as reported in previous work by our group [24,30]. 

Interestingly, the AMS programme reduced the use of antibiotics in the ICU but did not significantly lower overall consumption at the end of the follow-up. In addition, the number of patients receiving antibiotic therapy in the ICU during the point-prevalence surveys did not decrease throughout the study. In our opinion, these two observations reflect the necessity of using antibiotics in the ICU, given the clinical status of most of the patients admitted. As a result, achieving a statistically significant reduction in antibiotic use may be difficult. Consequently, the use of the narrowest spectrum antibiotic according to the microbiological culture results, or the reduction of antibiotic treatment duration, may be more appropriate outcomes for assessing the effect of AMS programmes in ICUs.

We also assessed ICU physicians’ adherence to guidelines, which often requires changes in prescribing behaviour. Involving local opinion leaders in AMS teams and increasing the perception of prescriber autonomy can facilitate the improvement in inappropriate prescribing of antibiotics [16,17,31,32]. We found that appropriateness of antibiotic prescribing increased significantly over time. In this respect, we believe that the inclusion of ICU leaders in the AMS team in 2016 helped us to achieve and maintain appropriate antibiotic use.

Finally, our results concur with those of previous studies [25,27,28,33] in demonstrating the safety of narrowing the antibiotic spectrum and shortening the duration of therapy in ICU patients.

To our knowledge, this is the first study to report a long-term reduction in ICU antibiotic use through the implementation of fully non-restrictive AMS strategies and also the first to provide evidence of improvements in prescribing quality over time. However, the study has several limitations. Firstly, the AMS programme was launched as part of an intervention to reduce endemic MDR-GNB. Therefore, as the study did not have a control group, infection control interventions may have confounded our results regarding the impact of the programme. However, as we have previously reported [24,30], many MDR-GNB in our ICU were resistant to carbapenems, piperacillin-tazobactam, and cefepime. Therefore, their effect on broad-spectrum antibiotic consumption is likely to have been small, given their lack of antimicrobial activity. Additionally, as the endemic rates of MDR-GNB decreased significantly after 2016, it could be speculated that this led to a reduction in the number of infections in the ICU. However, as can be seen from the point-prevalence surveys (Figure 3), the number of patients receiving antibiotic therapy did not decrease throughout the study period. Secondly, to prospectively assess adherence to guidelines and the number of patients receiving antibiotic therapy in the ICU, we relied on point-prevalence surveys that were conducted only once monthly. However, we were able to perform a large number of repeated measurements over a long follow-up period (seven years), which allowed us to estimate robust trends over the study period. Thirdly, because the study is based on stepwise multimodal interventions, we cannot measure the effect of each specific intervention, and so it is difficult establish which ones had the greatest effect on our findings. Finally, we did not analyse data on specific nosocomial infections during the study period, so it was not possible to compare trends.

## 4. Methods

### 4.1. Setting and Population

The AMS programme was implemented in three adult ICU wards with a total of 34 beds at Bellvitge University Hospital, a 700-bed teaching hospital located in the southern metropolitan area of Barcelona, Spain. The hospital is a referral centre for more than two million people requiring high-complexity care. The ICU is a mixed medical-surgical unit with an average of 1300 admissions per year, including abdominal surgery, neurosurgery, cancer, cardiothoracic surgery, and solid organ transplantation. Nursing staffing for the shift includes one nurse for every two beds and one assistant nurse for every four beds.

### 4.2. AMS Interventions

Between 2012 and 2016, several non-restrictive interventions were gradually implemented with the aim of reducing the use of broad-spectrum antibiotics (meropenem, imipenem, piperacillin-tazobactam, and cefepime) in the ICU. Table 1 shows the timeframe of the interventions implemented.

#### 4.2.1. Guidelines for Educational Support

In 2012, a multidisciplinary team including pharmacists, infectious diseases (ID) specialists, microbiologists, and ICU physicians developed a set of ICU-specific antimicrobial prescribing guidelines. The guidelines recommended empirical antibiotics and duration of therapy based on local bacterial susceptibility patterns and the site of infection. Educational rounds were conducted to engage ICU staff in the use of these guidelines. The guidelines were approved by the Hospital Infection Committee.

#### 4.2.2. Microbiological Support

Daily reporting of positive microbiological results and bacterial susceptibility patterns was already standard practice in the ICU at the start of the AMS programme. Results were available online and were also promptly reported to the ICU physicians by the ID consultant. 

#### 4.2.3. Optimisation of Antibiotic Dose

Therapeutic drug monitoring (TDM) and dose adjustment of glycopeptides and aminoglycosides have been routinely performed by pharmacists in the hospital since 2002. In 2016, routine TDM of ß-lactams was also included and the administration of ß-lactams as continuous infusion was encouraged. The pharmacokinetic/pharmacodynamic (PK/PD) target was to achieve a *f*C_ss_ 4 times higher than the minimum inhibitory concentration (MIC) of the pathogen during the entire dosing interval (100%T). AMS pharmacists individualised antibiotic dosing and administration schemes according to the PK of the ß-lactam.

#### 4.2.4. Setting up the AMS Team

In 2012, an AMS team consisting of the ID consultant and a clinical pharmacist started face-to-face prospective audits of all ongoing antibiotic prescriptions during daily ICU-patient rounds. The team aimed to provide feedback to clinicians and to make prescribing recommendations. In particular, they focused on narrowing the antibiotic spectrum according to the microbiological results and on reducing the number of days of therapy depending on the site of infection and based on local guidelines. Discontinuation of antibiotic therapy was also recommended when considered necessary. The patients’ treating clinicians were not obliged to follow the advice of the AMS team, and the final prescription decision was left to their discretion. 

Prior to starting the AMS programme, the ICU-ID consultant had already made recommendations to staff, but they had not been audited. No antibiotic prescription required approval prior to use in the ICU. To improve compliance with the AMS team’s recommendations, the ICU leaders from each ward, the head of the ICU, and a microbiologist joined the AMS team in 2016. This expanded AMS team met weekly to review all patients on antibiotics, with a particular focus on those on therapy for 7 or more days. Leaders of each ICU ward gave feedback on the group decision to their staff. Adherence to the decisions was not mandatory. 

#### 4.2.5. Point-Prevalence Audit Surveys and Feedback

In 2013, a monthly point-prevalence survey was introduced in order to assess the ICU staff’s compliance with local guidelines. This tool has demonstrated its usefulness for evaluating the appropriateness of antibiotic prescribing [34]. The survey collected information on the antimicrobials used on the day of the survey, start and end days of therapy, and the microbiological results of patient cultures. Prophylactic antibiotics or antibiotics started outside the ICU were also evaluated. Feedback and benchmarking of aggregate results were reported at ward level every 4 months and annually to encourage quality improvement in antibiotic prescribing.

#### 4.2.6. Electronic Support

In 2014, an automatic flag reminder showing the number of antibiotic days was added to the patient’s electronic prescription record. The number of antibiotic days was displayed and highlighted in red whenever any antibiotic was prescribed for 7 or more days.

#### 4.2.7. Financial Incentives

In 2016, the steering committee included an annual financial incentive for medical staff in the ICU if the annual adherence of antibiotic prescriptions to local antimicrobial guidelines reached 80% or more at ward level.

### 4.3. Outcomes and Data Collection

The main outcomes were: (1) *antibiotic use* measured as defined daily dose (DDD) per 100 patient-days (PD). DDD per 100 PD was calculated for total antibiotics and for each antibiotic class according to WHO and Anatomical Therapeutic Chemical definitions [35]. Data were collected on a 4-month basis from the pharmacy electronic dispensing programme and the standard hospital reporting system; (2) Appropriateness of antibiotic prescribing*,* which was considered if both the duration of therapy and the antibiotic spectrum were consistent with local guidelines and microbiological results. Appropriateness was measured by repeated point-prevalence surveys which were performed by the AMS team once a month (last working day). To obtain accurate information on duration of therapy and de-escalation of empirical therapy, the team reviewed patients’ electronic medical records. For critically ill patients with sepsis and negative cultures, maintaining broad-spectrum antibiotic therapy was considered an appropriate prescription.

The study recorded annual data on ICU length of stay, ICU mortality, and 30-day ICU readmissions to identify potential adverse outcomes during the implementation of the AMS. Data were obtained from the hospital’s standard reporting system.

### 4.4. Statistical Analysis

Time series analysis was used to describe total antibiotic use (overall and by class) from 2011 (pre-intervention year) to 2019. The trend and seasonality of each time series were assessed. The time points were summarised graphically using a local polynomial regression model (LOESS).

Data from monthly point-prevalence surveys were disaggregated per individual and calculated by month and year. As the point-prevalence surveys were not conducted in August 2013 and 2014 and July 2015, moving averages were imputed using four adjacent observations, with the lost observation as the centre of the period and linear weights. Trend and seasonality were also assessed, and time points were summarised graphically using the LOESS model.

A Poisson regression model was fitted to study the trends in the reduction of broad-spectrum antibiotic use and in the appropriateness of prescribing. The results were presented using incidence rate ratios (IRR) with a 95% confidence interval.

Differences in mean consumption between 2011 and 2019 were assessed using the Mann–Whitney U test, and differences in appropriateness of antibiotic prescribing between 2013 and 2019 were assessed using Pearson’s Chi-squared test. Data handling and analysis were performed with the R statistical programme, version 4.0 for Windows.

## 5. Conclusions

This study shows that a stepwise, multimodal, non-restrictive AMS programme achieved long-term reduction in the use of broad-spectrum antibiotics in the ICU and increased the appropriateness of antibiotic prescribing. 

## Figures and Tables

**Figure 1 antibiotics-13-00132-f001:**
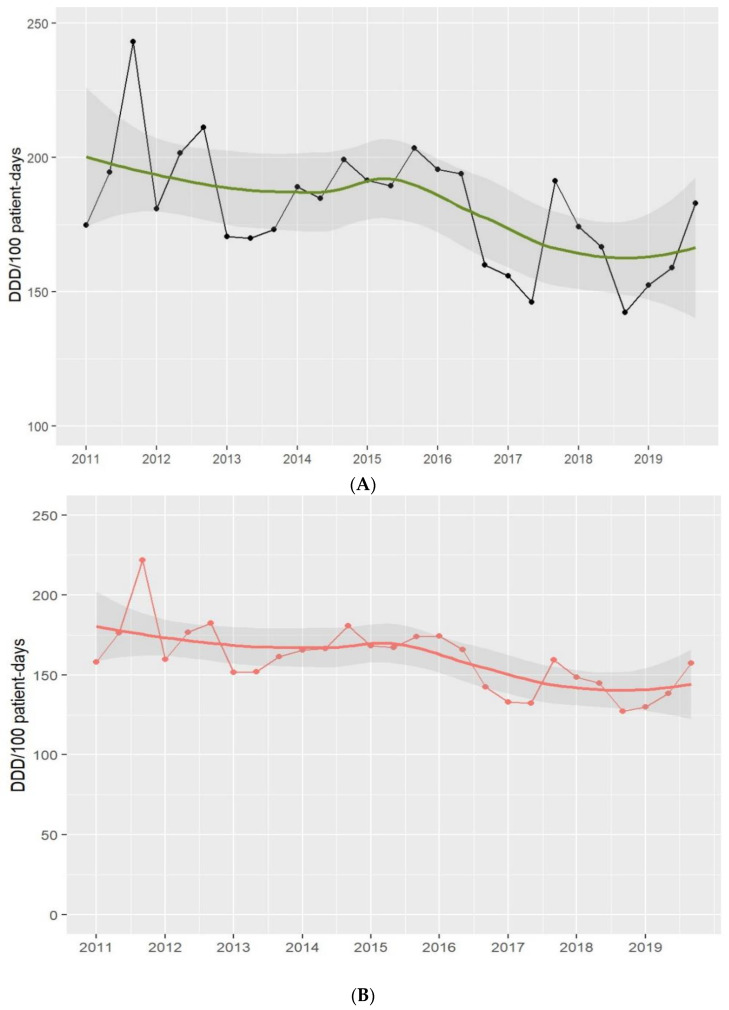
Antimicrobial (**A**) and antibiotic (**B**) use in the ICU. Period 2011–2019. Footnotes: Each dot represents the antimicrobial (black-(**A**)) and antibiotic (red-(**B**)) use in defined daily doses (DDD)/100 patient-days. The solid lines (green and red) are smooth curves obtained from the consumption data points using a local polynomial regression model (LOESS). The shaded areas represent the 95% confidence intervals of the LOESS curve.

**Figure 2 antibiotics-13-00132-f002:**
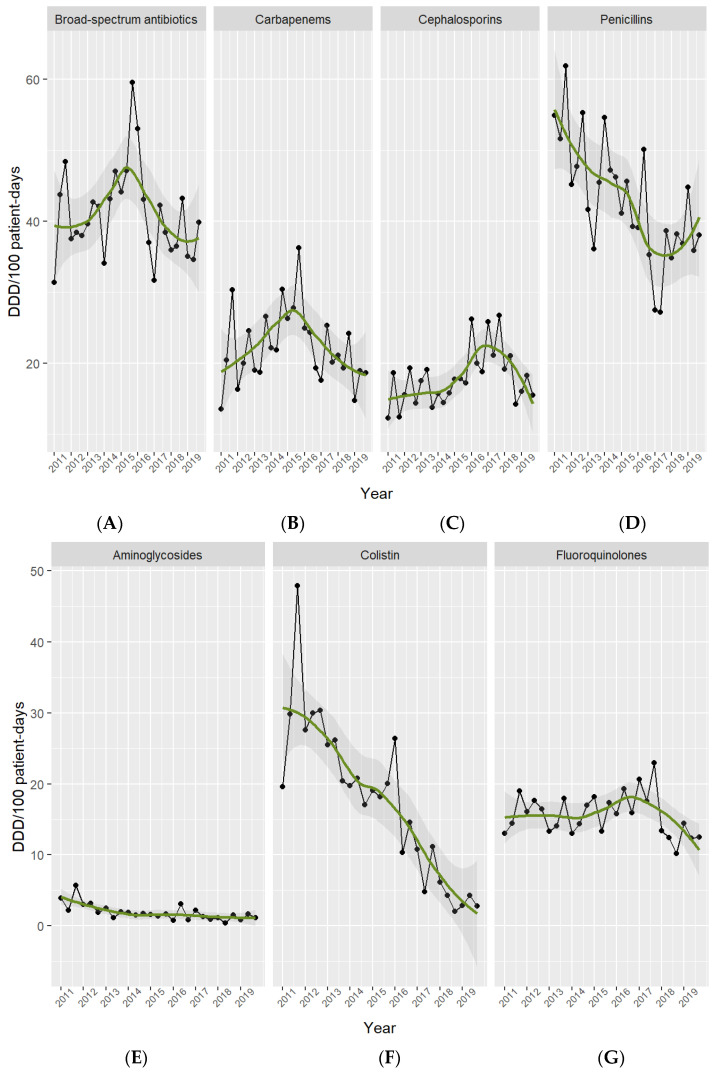
Antibiotic use per drug class in the ICU. Period 2011–2019. Footnotes: Broad-spectrum antibiotics (**A**) include meropenem, imipenem, piperacillin-tazobactam, and cefepime. Carbapenems (**B**) include meropenem, imipenem, and ertapenem. Cephalosporins (**C**) include cefuroxime, ceftriaxone, ceftazidime, and cefepime. Penicillins (**D**) include cloxacillin, penicillin G, ampicillin, amoxicillin-clavulanate, and piperacillin-tazobactam. Aminoglycosides (**E**) include amikacin, gentamicin, and tobramicin. Colistin (**F**). Fluoroquinolones (**G**) include ciprofloxacin, levofloxacin, and moxifloxacin.

**Figure 3 antibiotics-13-00132-f003:**
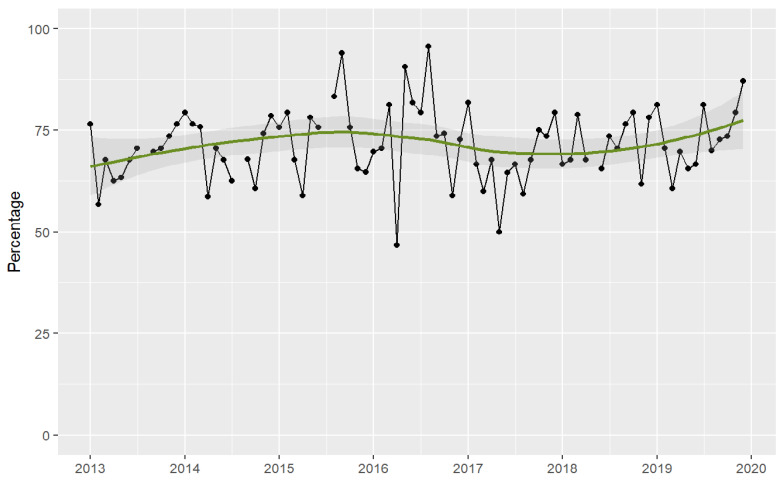
Patients receiving antimicrobial therapy in the ICU assessed by point-prevalence surveys. Period 2013–2019. Footnotes: Each dot represents the percentage of patients receiving antimicrobials. The solid lines (green) are smooth curves obtained from the patient data points using a local polynomial regression model (LOESS). The shaded areas represent the 95% confidence interval of the LOESS curve.

**Figure 4 antibiotics-13-00132-f004:**
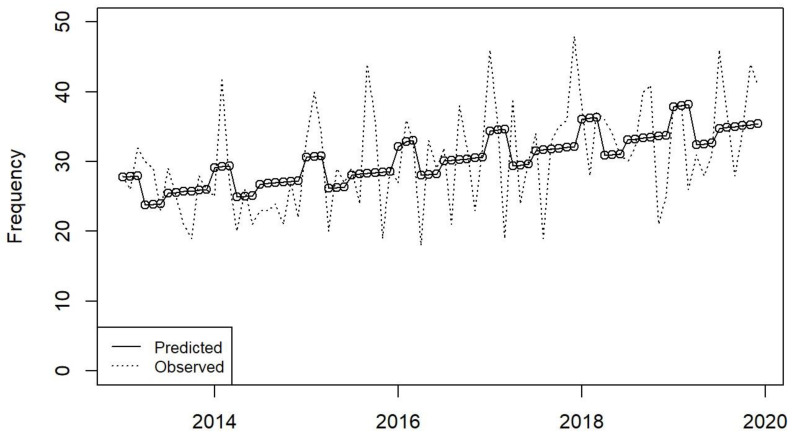
Moving average plot of appropriateness of antibiotic prescribing in the ICU in the period 2013–2019.

**Table 1 antibiotics-13-00132-t001:** Time frame of the interventions.

	2012	2013	2014	2015	2016	2017	2018	2019
Update of guidelines and education								
Microbiological support								
Setting up daily AMS team rounds								
Point-prevalence audit surveys and feedback								
ICU doctors added to the AMS team								
TDM and dose adjustment of ß-lactams								
Electronic flag reminder for prescription day								
Annual financial incentives								

AMS: antimicrobial stewardship; ICU: intensive care unit; TDM: therapeutic drug monitoring. The background colour indicates the period during which the interventions were implemented.

## Data Availability

Data contained within the article and Appendix A are available upon request from the corresponding author.

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
