# Peer review of "Long-Term Effects of a Stepwise, Multimodal, Non-Restrictive Antimicrobial Stewardship Programme for Reducing Broad-Spectrum Antibiotic Use in the ICU"

_antibiotics, 2024, doi:10.3390/antibiotics13020132_

Round 1
Reviewer 1 Report
Comments and Suggestions for Authors
I noticed the intensive efforts put over the years to implement proper use of antibiotics especially in the ICU. It's good to see a long term reduction in antibiotic use especially colistin.
My comments are:
1. The legends to the figures are not specified, further there is a capital F. On line106 need to be clarified. 2. The use of antimicrobial and antibiotics with separate curves red and black need to be clarified. I noticed both curves are identical but the numbers are lower in the curve with the red dots. 3. The lack of control group as you indicated in the limitations could affect the results. 4. The outcomes of the study in terms of length of stay, ICU mortalty and 30 days ICU readmission was not significant. If this study investigated these parameters on multidrug organisms such as Acinetobacter baumanni and others, it may give a different picture.
Comments on the Quality of English Language
NA
Author Response
Comments and suggestions: I noticed the intensive efforts put over the years to implement proper use of antibiotics especially in the ICU. It's good to see a long term reduction in antibiotic use especially colistin.
1.The legends to the figures are not specified, further there is a capital F. On line106 need to be clarified.
Response: Thank you for the valuable comment. We have added titles to all the figures (1, 2, 3 and 4) and clarified the legends. Highlighted in yellow in the main manuscript.
The capital F on line 106 is an error. We have deleted it.
- The use of antimicrobial and antibiotics with separate curves red and black need to be clarified. I noticed both curves are identical, but the numbers are lower in the curve with the red dots.
Response: Thank you for your comment on improving the display of our results. We have made the necessary changes to this figure. Figure 1 now includes subfigures F1A/F1B. Figure F1A shows the consumption of antimicrobials, including the use of antibiotics, antivirals, antifungals, and antiparasitics drugs in the ICU. F1B displays only the antibiotics, which were the main antimicrobials consumed in the ICU; hence, both curves were very similar. We have included this information in the text (Results section) for a better reader understanding (see page 3, lines 112-116). Additionally, we have clarified the footnotes. (see page 4, lines 137-141)
“Antimicrobial consumption, including the antibiotic, antifungal, antiviral and antiparasitic use, fell from 204.1 in 2011 to 164.7 DDD per 100 PD in 2019 [absolute difference, -39.4 (19%), 95% CI -113.54 to 34.74; p= 0.18]. The decrease was mainly driven by antibiotic consumption, which fell from 185.4 in 2011 to 141.9 DDD per 100 PD in 2019 [absolute difference, -43.5 (23%), 95% CI -100.73 to 13.73; p= 0.13].”
“Footnotes: Each dot represents the antimicrobial (black-Figure 1A) and antibiotic (red-Figure 1B) use in defined daily doses (DDD)/100 patient-days. The solid lines (green and red) are smooth curves obtained from the consumption data points using a local polynomial regression model (LOESS). The shaded areas represent the 95% confidence intervals of the LOESS curve”.
- The lack of control group as you indicated in the limitations could affect the results.
Response: Thank you for the pertinent comment. This allowed us to justify our thinking regarding this limitation. The following information has been added to the text (see page 7 lines 231-241)
The AMS programme was launched as part of an intervention to reduce endemic MDR-GNB. Therefore, infection control interventions may have confounded our results regarding the impact of the programme. However, as we have previously reported1, many of the MDR-GNBs in our ICU were resistant to carbapenems, piperacillin-tazobactam, and cefepime. Therefore, their effect on broad-spectrum antibiotic consumption is likely to have been small, given their lack of antimicrobial activity, especially in the early years of the programme when their consumption increased, as shown in Figure 2. Additionally, as the endemic rates of MDR-GNBs decreased significantly after 20162, it could be speculated that this could lead to a reduction in the number of infections in the ICU. However, as can be seen from the point prevalence surveys (Figure 3), the number of patients receiving antibiotic therapy did not decrease throughout the study period. Unfortunately, we do not have data on all patients receiving antibiotics in the ICU during the study period. The point prevalence surveys are a “proxy”. However, we were able to perform a large number of repeated measurements over seven years, which allowed us to estimate robust trends over the study period.
It should be noted that the infection control interventions did not change throughout the study period.
1.Gavaldà L, Soriano AM, Cámara J, et al. Control of endemic extensively drug-resistant Acinetobacter baumannii with a cohorting policy and cleaning procedures based on the 1 room, 1 wipe approach. Am J Infect Control. 2016; 44:520-4.
2.Shaw E, Gavaldà L, Càmara J, et al. Control of endemic multidrug-resistant Gram-negative bacteria after removal of sinks and implementing a new water-safe policy in an intensive care unit. J Hosp Infect. 2018; 98:275-281.
4. The outcomes of the study in terms of length of stay, ICU mortality and 30 days ICU readmission was not significant. If this study investigated these parameters on multidrug organisms such as Acinetobacter baumanni and others, it may give a different picture.
Response: Thank you, we appreciate this interesting question.
A. baumanni and other MDR-GNBs cause difficult-to-treat infections; hence, all the outcomes mentioned above are expected to worsen. However, previous research by our group showed that despite the introduction of XDR A. baumannii in our ICU in 19921, there has been a significant increase in the number of new cases since 2011. To address this problem, we implemented a series of interventions in 2011, with a particular focus on cleaning policies2. The results of these interventions were successful in controlling A baumannii; however, they did not control other MDR-GNB, specifically, K. pneumoniae producing BLEE and/or carbapenemase and XDR-P aeruginosa producing carbapenemase or not. This new scenario led our infection control team to remove sinks from ICU rooms 3.
In this context, the AMS programme presented in this article includes the outcomes of patients in our ICU from 2011 to 2019. Therefore, in response to your question, we would say that we did not observe any relevant differences in these results.
References:
1.Corbella X, Pujol M, Ayats J, et al. Relevance of digestive tract colonization in the epidemiology of nosocomial infections due to multiresistant Acinetobacter baumannii. Clin Infect Dis 1996; 23: 329-34.
2.Gavaldà L, Soriano AM, Cámara J, et al. Control of endemic extensively drug-resistant Acinetobacter baumannii with a cohorting policy and cleaning procedures based on the 1 room, 1 wipe approach. Am J Infect Control. 2016; 44:520-4.
3.Shaw E, Gavaldà L, Camara J, et al. Control of endemic multidrug-resistant Gram-negative bacteria after removal of sinks and implementing a new water-safe policy in an intensive care unit. J Hosp Infect. 2018; 98:275-281.
Reviewer 2 Report
Comments and Suggestions for Authors
This is a good study on an important subject. It is recommended to also show p values in tables.
Author Response
Comments and suggestions: This is a good study on an important subject. It is recommended to also show p values in tables.
Response: Thank you for the comment. Notably, as is shown in the attached Table S1, the statistical tests indicate statistically significant findings for all variables. However, we would like to express our concern about the lack of clinical relevance despite these statistical significances. The large sample sizes allow that small differences to reach statistical significance.
In this regard, the data collected for the study are annual aggregated data that we obtained from the standard hospital reporting system. Therefore, because we did not have patient-level information, the p-value statistics for these relevant outcomes (the length of stay, ICU mortality, and 30-day readmissions) cannot be adjusted for confounding. Furthermore, if we look at the raw results of the different years included in the study, they were not too different in 2019 from those in 2011 (the year prior to the implementation of the AMS programme). Therefore, since Table S1 presents unadjusted information, we prefer not to include the p-value as it may be confusing for the reader. We have added a footnote to the Table S1.
" Aggregated data from the hospital’s standard reporting system"
In Table 2 of the Supplementary Appendix (Table 3 in the new manuscript version), and as suggested by the reviewer, the p-values have been included. In addition, we have added this information in the text (see page 5, lines 159-161).
“”The number of patients receiving antibiotics in the monthly surveys did not decrease throughout the study period. There was a trend towards an increase in the number of patients with antibiotic treatment over time. See Figure 3 and Table S3 in the Supplementary Appendix.
Note: Please, find Tables with P-value attached.

Reviewer 3 Report
Comments and Suggestions for Authors
Long-term effects of a stepwise, multimodal, non-restrictive antimicrobial stewardship programme for reducing broad-spectrum antibiotic use in the ICU
Dear author and editor:
The study talked about the effect of multimodal, non-restrictive AMS programme in reducing the antibiotic use and improved the antibiotic prescribing.
The article could be published after a revision
I have some comment on it:
· More detailed information have to be added to the introduction with more citation.
· The author could also talked about resistance and multidrug resistance infection as an increasing problem in the introduction section.
· The figure number should be followed by the legend directly.
· A title should be added to the each figure. And you could add Fig 1 A , 1B for multiple sub figures.
· According to your results. Did AMS program reduce the MDR spread?
Thank you very much. Best regards
Author Response
Comments and suggestions: The article could be published after a revision. I have some comment on it:
1.More detailed information have to be added to the introduction with more citation. The author could also talked about resistance and multidrug resistance infection as an increasing problem in the introduction section.
Response: Thank you for the comments. This has allowed us to improve the Introduction. The current Introduction has been extended from 302 to 517 words and includes 24 references (six more references than in the previous version).
As suggested, the Introduction section has been improved to provide information on the importance of multidrug-resistant infections. The following information has been added to the text (see page 2 lines 58-65)
“Antimicrobial resistance (AMR) is a priority issue for the World Health Organization. It is responsible for an estimated 700,000 deaths annually due to hard-to-treat infections. Additionally, misuse and overuse of antimicrobial agents in human medicine and food production chains have been identified as main drivers of AMR.
Antimicrobial stewardship (AMS) programmes have a key role to play in reducing AMR and tackling the emergence and spread of multidrug-resistant (MDR) bacteria. It is essential to implement effective measures able to reduce unnecessary and inappropriate use of antimicrobial agents without negatively affecting patient outcomes”.
2.The figure number should be followed by the legend directly. A title should be added to the each figure. And you could add Fig 1 A, 1B for multiple sub figures.
Response: Thank you for the comment. We have added titles to all the figures (1, 2, 3, and 4) and clarified the legends. Following your suggestions, Figures 1 and 2 includes sub-figures. Highlighted in yellow in the main manuscript.
- According to your results. Did AMS program reduce the MDR spread?
Response: Thank you for your kind comments. We appreciate your interesting question.
It is important to note that this AMS programme was initiated as part of wider interventions to address the problem of endemic MDR-GNB, specifically, K. pneumoniae producing BLEE and/or carbapenemase and XDR-P aeruginosa producing carbapenemase, which we encountered in our ICU. As other measures have been implemented since 2011 to reduce the spread of MDR-BGN, such as cleaning policies and removing sinks from ICU rooms, we could not specifically assess the impact of the AMS programme on reducing the spread of MDR-GNB. We can tell you that we do not currently have any relevant problem with MDR-BGN. Therefore, we strongly believe that to improve the impact of AMS programmes on the spread of MDR-GNB in settings such as the ICU, they need to be accompanied by other measures.
Note: Please, find the new version of Introduction attached.

Reviewer 4 Report
Comments and Suggestions for Authors
In the introduction section, authors can add some references regarding healthcare workers' knowledge and attitudes regarding antibiotic use
Every figure needs a title and a legend
Methodology must be better pesented
Comments on the Quality of English LanguageVery interesting manuscript
Author Response
Comments and suggestions: Very interesting manuscript
1.In the introduction section, authors can add some references regarding healthcare workers' knowledge and attitudes regarding antibiotic use.
Response: Thank you for the valuable comments. This has allowed us to improve the Introduction section.
The current Introduction has been extended from 302 to 517 words and includes 24 references (six more references than in the previous version).
As suggested, the Introduction section has been improved to provide information on the attitudes of health workers towards antibiotic prescribing. The following information has been added to the text (see page 2 lines 82-96)
“Non-restrictive AMS policies (also known as persuasive strategies) are the ones that physicians prefer for reducing antibiotic use [14-15]. A recent cross-sectional survey conducted in France, which explored the preferred AMS strategies of hospital prescribers found that prescribers considered educational actions to be more useful than restrictive ones. They felt that restrictive interventions undermined their clinical autonomy [16].
Studies investigating the social and cultural determinants of the prescribing behaviour of physicians in the hospital setting have identified some physicians’ prescribing beliefs, such as that senior doctors often base their decisions on personal knowledge and experience despite not having specific knowledge about antimicrobial use. In addition, junior doctors tended to base their antibiotic prescribing on the senior’s decision [16-18]. Consequently, involving senior staff in the development of local AMS interventions may promote changes in prescribing habits, especially among younger physicians, and help address inappropriate antibiotic prescribing.”
2.Every figure needs a title and a legend
Response: Thank you for the comments. We have added titles to all the figures (1, 2, 3, and 4) and clarified the legends. Highlighted in yellow in the main manuscript.
3.Methodology must be better presented
Response: Thank you for the comment. Unfortunately, we are unable to fully understand what you mean by “better presented”. Please provide us with additional information regarding this.
In this regard, with the aim of clarifying some sections of the methodology, we have added short sentences to various paragraphs of the text. These have been highlighted in yellow in the main manuscript.
Note: Please, find the new version of Introduction attached

Round 2
Reviewer 1 Report
Comments and Suggestions for Authors
The new version with the corrections is accepted